# Uncovering Novel Pre-Treatment Molecular Biomarkers for Anti-TNF Therapeutic Response in Patients with Crohn’s Disease

**DOI:** 10.3390/jfb13020036

**Published:** 2022-03-30

**Authors:** Min Seob Kwak, Jae Myung Cha, Jung Won Jeon, Jin Young Yoon, Su Bee Park

**Affiliations:** Department of Internal Medicine, Kyung Hee University Hospital at Gangdong, Kyung Hee University College of Medicine, Seoul 05278, Korea; a9391@hanmail.net (J.M.C.); 4528359@gmail.com (J.W.J.); miju8339@naver.com (J.Y.Y.); j-miju@hanamil.net (S.B.P.)

**Keywords:** Crohn’s disease, anti-TNF, response, biomarker, transcriptome, KAT2B, IL-17

## Abstract

Neutralising monoclonal antibodies for tumour necrosis factor (TNF) has been widely used to treat Crohn’s disease (CD) in clinical practice. However, differential individual response necessitates a therapeutic response assessment of anti-TNF agents in CD patients for optimizing therapeutic strategy. We aimed to predict anti-TNF therapy response in CD patients using transcriptome analyses. Transcriptome analyses were performed using data from the Gene Expression Omnibus, GeneCards, and Human Protein Atlas databases. The significantly mitigated biological functions associated with anti-TNF therapy resistance in CD patients encompassed immune pathways, including Interleukin-17 (IL-17) signaling, cytokine-cytokine receptor interaction, and rheumatoid arthritis. The scores of immune cell markers, including neutrophils, monocytes, and macrophages/monocytes were also significantly decreased in non-responders compared with that measured in anti-TNF therapy responders. The *KAT2B* gene, associated with IL-17 cytokine mediated neutrophil mobilization and activation, was significantly under-expressed in both tissue and peripheral blood mononuclear cells (PBMCs) in anti-TNF therapy-resistant CD patients. The reduced expression of several pro-inflammatory cytokines due to down-regulated IL-17 signaling, is suggestive of the primary non-response to anti-TNF agents in CD patients. Furthermore, the PBMC *KAT2B* gene signature may be a promising pre-treatment prognostic biomarker for anti-TNF drug response in CD patients.

## 1. Introduction

Crohn’s disease (CD) is one of the major chronic relapsing disorders associated with inflammatory bowel disease (IBD) that can affect any part of the gastrointestinal tract and thus, greatly influence the quality of life [1,2,3]. Recent evidence suggests that dysregulated responses attributable to the molecular alterations from the interplay of genetics, gut microbiome, and environmental factors considerably influence the flares of CD [4,5].

Clinically, the therapeutic goals in CD aim to induce mucosal healing of the intestine and sustain remission. At present, the tumour necrosis factor α (TNF-α) that induces intestinal infiltration of inflammatory cells as a major pathological cytokine has been validated as a drug target in IBD. Accordingly, a blockade of its signal by monoclonal antibodies has become the most widely used therapeutic strategy for the treatment of steroid-dependent/refractory CD patients [6].

Unfortunately, studies have documented that up to 40% of refractory CD patients primarily do not respond to anti-TNF drugs. Moreover, nearly half of the patients who benefit from the treatment eventually become non-responsive within the first year, necessitating subsequent therapy change [7,8]. These findings indicate that the anti-TNF resistance in CD patients may arise from a variety of underlying biological mechanisms. To date, few methods based solely on clinical parameters and disease activity scores have been suggested to study the response to anti-TNF therapy in CD. Furthermore, they do not yield the required efficacy, even for a baseline primary response [9]. Therefore, a clear understanding of their resistance mechanisms and the pre-treatment assessment of anti-TNF therapy responsiveness in CD are critical for patients who may be susceptible to severe adverse effects of the drug that they may not benefit from, their families, and healthcare systems that endure the additional economic liability due to poor disease outcome. 

Here, we aimed to elucidate the molecular mechanisms responsible for anti-TNF therapy resistance in CD patients. Furthermore, we attempted to identify the candidate blood transcriptomic signatures to predict the pre-treatment drug response for anti-TNF agents in CD patients.

## 2. Materials and Methods

### 2.1. Description of Datasets

The datasets used in this study were retrieved from the Gene Expression Omnibus database (http://www.ncbi.nlm.nih.gov/geo/ (accessed on 1 November 2021)) of the National Center for Biotechnology Information. We analysed the gene expression profiles from the ileo-colonic specimen of 36 CD patients from the GSE16879 dataset that includes information about CD therapy responses from 19 responders and 17 primary non-responders for anti-TNF drugs. To further explore the efficacy of potential blood biomarkers, the expression of some of the putative key genes in the pre-treatment assessment of anti-TNF resistance in CD was analysed from the data in the GSE42296 dataset, which included the microarray expression data from peripheral blood mononuclear cells (PBMCs) in a total of 14 responders and 6 primary non-responders for anti-TNF therapy. All patients received infliximab caused by the resistance to corticosteroids and/or immunosuppressants, and their samples were obtained before and after therapy. Detailed information of the datasets can be found in Appendix A.

### 2.2. Biological Function Analysis

First, we performed the normalisation of each dataset using the robust multi-array average method. After log2-transformations, linear modelling frameworks and empirical Bayes approaches were employed to identify the gene or microenvironmental features that were associated with the anti-TNF therapy resistance in CD patients. To discover the dysregulated pathways associated with the anti-TNF therapy resistance in CD, we performed a differential expression analysis between responders and non-responders. The *p*-values were adjusted by weighting the Benjamini and Hochberg false discovery rate control procedure [10]. The genes satisfying both the selection criteria of *p*-value < 0.05 and |logFC| ≥ 0.5 were selected as candidate genes. The gSOAP R package was used to perform the over-representation analysis of the gene set in the Kyoto Encyclopedia of Genes and Genomes (KEGG) database [11]. For functional enrichment analyses, the biological processes and molecular functions related to the genes of interest between the two groups were assessed by the clusterProfiler [12] R package through the KEGG database. 

To improve the interpretation of intermolecular interaction, analysis using GeneAnalytics, a gene set analysis tool (https://www.genecards.org/ (accessed on 1 December 2021)) was also utilised in accordance with the existing literature. Additionally, we analysed the cellular expression of proteins corresponding to candidate biomarker genes in the Human Protein Atlas (HPA, https://www.proteinatlas.org/ (accessed on 15 December 2021)) [13,14,15,16]. 

### 2.3. Microenvironmental Feature Evaluation

For estimating the abundance of immune and stromal signatures, the Microenvironment Cell Population (MCP)-counter, a gene expression-based computational approach, was employed. 

The signatures were discovered using the specific cut-offs through transcriptomic analysis and underwent an additional selection process. A sub-signature with strong inter-marker correlation was kept following hierarchical clustering of the whole signature transcriptomic markers.

Then, the in silico simulated mixtures were computed as follows: firstly, weights for all included populations were chosen randomly. Pure transcriptomic profiles for all populations were computed with the expression of all genes being the mean expression over all the corresponding samples in the datasets. The absolute abundance scores for fibroblasts, neutrophils, endothelial cells, monocytic lineage cells, B lineage cells, NK cells, myeloid dendritic cells, cytotoxic lymphocytes, CD8^+^ T cells, and T cells were calculated; the scores were defined as mean log2 expressions [17]. 

### 2.4. Statistical Analyses

The data were analysed using two-tailed tests; *p*-values ≤ 0.05 were considered statistically significant. All data processing and analyses were performed using R (version 4.0.5) and Python (version 3.7.1) scripts. The analyses were run on a server with a 2 × Six-Core Intel Xeon processor, two-GPU Nvidia TITAN X, and 128 GB memory. This study was evaluated and approved by the Institutional Review Board of the Kyung Hee University Hospital at Gangdong, Seoul, Republic of Korea (KHNMC IRB 2022-01-016).

## 3. Results

### 3.1. Identification of the Target Gene Signatures Attributed to Anti-TNF Therapy Resistance 

We identified 312 differentially expressed genes (DEGs), of which 17 DEGs were up regulated, while the remaining 295 were down regulated in the anti-TNF therapy non-responder groups as compared with those in the anti-TNF responder groups. To gain crucial insights into the biological processes regulated by the target DEGs, their functional annotation was performed. For the top five most significant KEGG terms, we utilized an over-representation approach of the gene set to demonstrate that their functions could be classified into three groups (Figure 1). The first cluster group represented the ‘Rheumatoid arthritis pathway’, ‘Interleukin-17 (IL-17) signaling pathway’, ‘Cytokine-cytokine receptor interaction pathway’, ‘Focal adhesion pathway’, and ‘Osteoclast differentiation pathway’. The second cluster group included the ‘TNF signaling pathway’ and ‘Toll-like receptor signaling pathway’. Furthermore, the top two enriched pathways for genes clustered in the third cluster group were pathways involving ‘Renal cell carcinoma’ and ‘Pancreatic cancer’ (Figure 1).

### 3.2. Functional Enrichment Analysis of the Top DEGs Associated with Anti-TNF Therapy Resistance

In Figure 2, The functional enrichment analysis of DEGs associated with resistance to an anti-TNF agent in the top five pathways, including ‘Rheumatoid arthritis’, ‘IL-17 signaling pathway’, ‘Cytokine-cytokine receptor interaction’, ‘Focal adhesion’, and ‘Osteoclast differentiation’, demonstrates that majority of the genes associated with therapy resistance were predominantly down-regulated (Figure 2). For the predominant functional modules, the top interacting DEGs to anti-TNF drug between non-responders and responders were mainly enriched in immune-related functions, including the ‘IL-17 signaling’, ‘Rheumatoid arthritis’, and ‘Cytokine-cytokine receptor interaction’ pathways (Figure 3). Furthermore, the resultant DEGs were also functionally annotated using the KEGG pathway analysis, with the ‘Cytokine-cytokine receptor interaction’ within the ‘Chemokine signaling pathway’ and ‘IL-17 signaling pathway’ being the most relevant term to intestinal inflammation (Figure 4 and Appendix A). The up- or down-regulation of *IL-17* genes (*IL-17A*, IL-17B, *IL-17D*, and *IL-17F*) resulted in decreased expression of many cytokines and chemokines, which are associated with autoimmunity and immune cell functions (Appendix A). Furthermore, most of the genes involved in the “Chemokine signaling pathway” initiated by various immune cells were down-regulated in anti-TNF CD therapy in non-responders as compared to that in therapy responders (Figure 4).

### 3.3. Tissue-Microenvironment Landscape Associated with Anti-TNF Therapy Resistance

Herein, we sought to explore possible molecular mechanisms through immune cell infiltration during the aetiology of resistance to anti-TNF therapy. Accordingly, the MCP-counter method was employed to analyse the expression scores, which were significantly lower in fibroblasts (*p* < 0.001), neutrophils (*p* < 0.001), endothelial cells (*p* = 0.001), monocytes (*p* = 0.004), and macrophages/monocytes (*p* = 0.004) (Table 1) of anti-TNF-resistant patients compared to that in those of anti-TNF therapy-responding patients. Moreover, the analyses further revealed that other immune cell subtypes were also less abundant in the patients who did not respond to anti-TNF therapy as compared to those who did respond, although this finding was not statistical significance (Table 1).

### 3.4. Development of Biomarkers for Anti-TNF Therapy Resistance

To explore candidate blood biomarkers for predicting primary non-response to anti-TNF therapy, a differential expression analysis was performed comparing the expression data from PBMCs of anti-TNF-resistant patients vs. CD patients, who responded to the anti-TNF agent. Subsequently, the list of genes differentially expressed in PBMCs was examined for significant overlap with the list of genes previously identified as differentially expressed in tissue samples of CD patients using the GSE16879 dataset. The results revealed that the expression of nine genes, namely, *CBR4*, *ACAD10*, *BAIAP3*, *BMP6*, *DDX11L2*, *DYNLL2*, *KAT2B*, *KLF9*, and *SPTSSB*, in both PBMCs and tissue samples were frequently altered at baseline in anti-TNF therapy responders and non-responders. Subsequent filtering, taking into consideration the absolute log2 fold change and the level of statistical significance in the expression levels of these genes between the two groups, resulted in a list of two genes (*DDX11L2*, and *KAT2B*) identified as key target genes that could effectively differentiate non-responders from responders for anti-TNF therapy at baseline. Further analysis using the GeneCards Suite’s Gene Analytics tool revealed *KAT2B* as the gene with the most super-enhancer identifiers associated with IL-17 function (Table 2). We further explored the association between various immune cell types and the KAT2B protein using the HPA database and demonstrated that neutrophils were the immune cells with the most abundant expression of *KAT2B* transcripts, as indicated by the quantitative results (Appendix A).

## 4. Discussion

To the best of our knowledge, this is the first study that establishes the putative biological mechanisms underlying anti-TNF therapy resistance in CD patients. Our results demonstrate that decreased immune-related pathways at baseline suggests a core pathological role for anti-TNF resistance in CD and reports the usefulness of the *KAT2B* gene in PBMCs as a prognostic biomarker for anti-TNF therapy resistance in CD.

Although anti-TNF drugs have become the mainstay in CD treatment, their mechanism of action is still under debate. They have been reported to deploy more complex mechanisms, including shifts in immune cell populations [18,19] and prevention of epithelial barrier dysfunction [20,21] beyond the simple blockade of the TNF cytokine, which may be harmful and cause colorectal inflammation due to excessive production by direct neutralisation [22]. Therefore, considerable heterogeneity of the anti-TNF responsiveness in CD patients may be attributed to the fact that these drugs may induce several of these mechanisms at any given point, thereby necessitating the development of an individualised approach for anti-TNF therapy application depending on the active mechanism.

Several biologic markers such as serum cytokine levels (TNF and IL-6), C-reactive protein, anti-neutrophil cytoplasmic antibody, anti-Saccharomyces cerevisiae antibody, and anti-drug antibody have been studied [23,24,25,26]. However, they could not unmask the mechanisms represented due to the nature of observational studies; we therefore have not yet seen data convincing enough to use this routinely in clinical practice.

Only a few studies have used this approach, although the identification of transcriptomic biomarkers using DEGs in CD could be useful to elucidate the mechanism of the phenotype and to identify the patients who are less likely to respond in early stages or even before initiation for anti-TNF therapy [27,28,29]. Unfortunately, most of these studies used either tissue or blood samples, while we use the multi-omics data from tissue and blood samples to characterise CD patients by underlying gene transcription signatures and identify DEGs impacting biological pathways in anti-TNF resistance.

Our results show that the down-regulated DEGs in the anti-TNF therapy-resistant CD patients vs. CD patients responding to the anti-TNF drug were clustered with the highest scoring in immune-related pathways, such as ‘IL-17 signaling’, ‘Cytokine-cytokine receptor interaction’, and ‘Rheumatoid arthritis’. Various cytokines and chemokines play a major role in the regulation of mucosal immunity in the intestine by promoting leukocyte recruitment/migration to inflammatory sites, ultimately leading to the damage and destruction of the intestinal tissue of CD patients. Among these cytokines, IL-17 is involved in the immune response against extracellular pathogens through the regulation of the intestinal barrier and maintenance of intestinal homeostasis [30]. Moreover, IL-17 together with TGF-β, IL-6, IL-1β, IL-21, and IL-23 are produced by the activation of Th17 cells, a third subset of Th lymphocytes that are mostly located in the lamina propria of the gastrointestinal wall. Although not directly related with TNF-α, these cytokines and chemokines are responsible for triggering and amplifying the inflammatory process in CD [31,32,33], potentially indicating their role as crucial drivers of anti-TNF therapy resistance. Previous studies have reported that colitis is associated with an increase in IL-17A subunit levels, while IL-17F subunit expression is inversely correlated with gut inflammation [34], which subsequently prompts neutrophil recruitment into the ileum [35]. Interestingly, the present study also illustrates that significantly elevated *IL-17F* and reduced *IL-17A* expression levels are followed by the down-regulation of many proinflammatory cytokine and chemokine genes (*CXCL-5*, *CXCL-8*, *G-CSF*, *CCL-2*, *COX-2*, and *IL-6*) in the tissues of anti-TNF therapy-resistant CD patients as compared with those noted in CD patients that respond to the therapy. These findings implicate that the dysregulation of proinflammatory genes in the patients with anti-TNF therapy resistance can negatively affect the recruitment and activation of neutrophils.

In light of previous evidence, the results of our study assert that anti-TNF therapy resistance in CD patients may be attributed to the low immune cell activation due to differential gene expression status of *IL-17A/IL-17F* between the responders and non-responders to anti-TNF agents. Therefore, we additionally performed microenvironmental cell profile analysis; as expected, the results revealed significantly lower scores of several immune-cell subsets: neutrophils, monocytes, and macrophages/monocytes in anti-TNF therapy-resistant patients compared to those in the CD patients that responded to the drug.

Our in silico findings alone, without experimental validation, do not completely explain the mechanism underlying the responsiveness of patients to anti-TNF-α therapy in CD. However, our findings are in line with cumulative evidence and may have relevant implications for the prediction of therapeutic response to anti-TNF agents. Furthermore, our findings suggest that immune phenotyping may facilitate a more accurate stratification of CD patients during selection for anti-TNF therapy.

Next, the detection of gene biomarkers with the potential capability of differentiating non-responders from responders to anti-TNF therapy is clearly an unmet need in the management of CD at present. Several clinical biomarkers, which serve as reproducible as well as quantitative tools to assess therapeutic efficacy during anti-TNF therapy, have been previously investigated in IBD patients, including C-reactive proteins, faecal calprotectins, and anti-TNF drug levels [9]. However, at present, no clinical predictors exist that predict the anti-TNF therapy response at baseline prior to its application, even though preemptive prediction can serve to be more meaningful in the individualised management of CD.

In this study, we have found that *KAT2B* was commonly down-regulated in both PBMCs and the intestinal tissues of anti-TNF therapy non-responders as compared with those in the CD patients that respond to the anti-TNF therapy. Hence, we rationalised that KAT2B could serve as a potential biomarker based on the literature and protein expression evidence from the HPA. KAT2B, the epigenetic regulator, is a lysine acetyltransferase that is down-regulated in the inflamed colonic tissue of CD patients [36]. So far, few studies have reported the role of the *KAT2B* gene in the prediction of anti-TNF resistance in CD [37,38]. Mesko et al. have reported that the gene panel including *KAT2B* may be useful for the prediction of anti-TNF responsiveness in CD [37]. Furthermore, a recent integrative transcriptomic and genomic study by Gorenjak et al. has suggested remarkable potential of *KAT2B* as a prognostic biomarker for anti-TNF therapy responsiveness in patients with CD [38]. However, the studies were investigational in nature and did not stress the biological interpretation of the molecular profiling data. In this study, we overcame this shortcoming by presenting a more detailed interpretation of the possible signaling mechanisms responsible for the primary non-response to anti-TNF therapy. Thus, our systematic analysis presents higher probability for identifying an effective prognostic biomarker, thereby laying the foundations for the advancement of personalised therapy in CD. 

Our study has certain limitations. First, the sample size of the analysis was relatively small. Second, we focused on analysing the gene expression profiles from both serum and tissue samples for identifying candidate biomarkers in CD patients; however, we did not use the paired transcriptomic datasets from the same sample group. Until now, few databases have produced considerable multi-omics data from one sample in CD. Third is the lack of clinical information, such as age, gender, smoking status, and co-prescribed drugs. Thus, prospective studies are required to verify our findings.

## 5. Conclusions

In conclusion, our results reveal that the activity of IL-17 in intestinal tissues of CD patients at baseline may be heterogeneous, resulting in a diverse immune microenvironment, which in turn, promotes anti-TNF drug resistance. Moreover, we have shown that the expression of *KAT2B* in the PBMCs of CD patients could serve as a prognostic and immunological biomarker for anti-TNF therapy resistance. However, detailed experimental studies are essential to validate our findings.

## Figures and Tables

**Figure 1 jfb-13-00036-f001:**
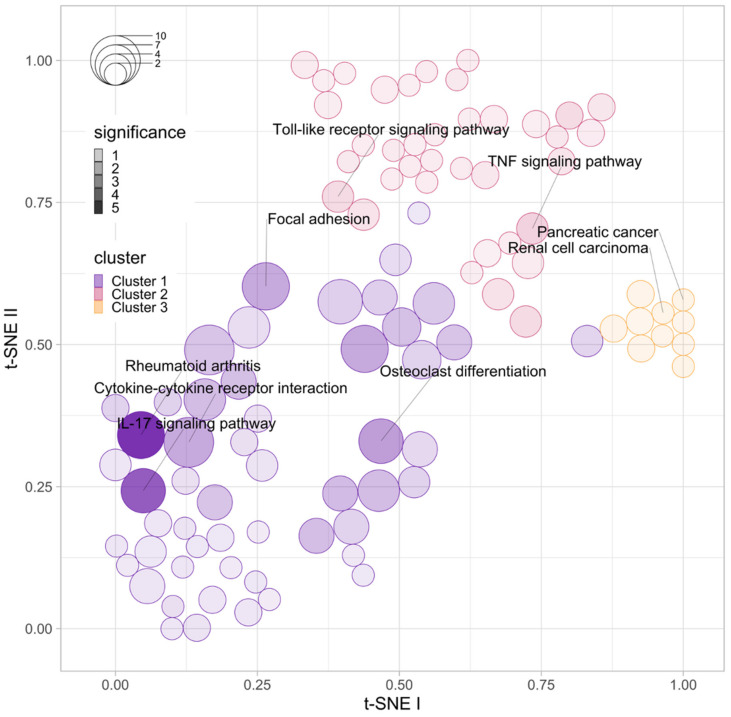
Functional enrichment analysis based on KEGG pathways and differentially expressed genes associated with anti-TNF therapy resistance in Crohn’s disease. Circle size and colour indicate the number of genes enriched in a specific pathway and their significance, respectively.

**Figure 2 jfb-13-00036-f002:**
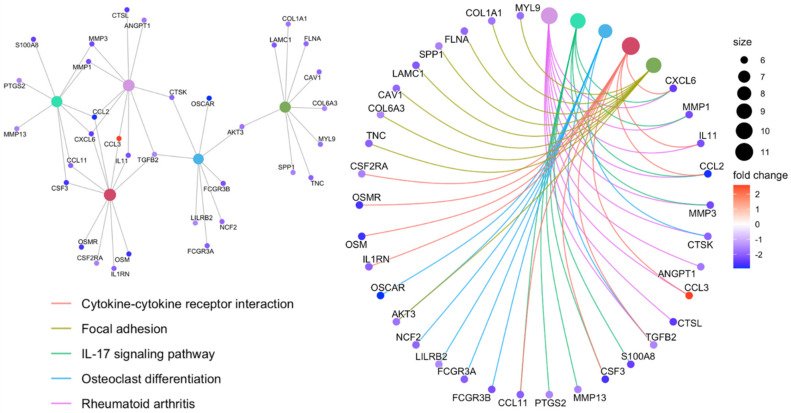
Commonly deregulated targeted genes associated with anti-TNF therapy resistance in Crohn’s disease and associated pathways in cluster 1.

**Figure 3 jfb-13-00036-f003:**
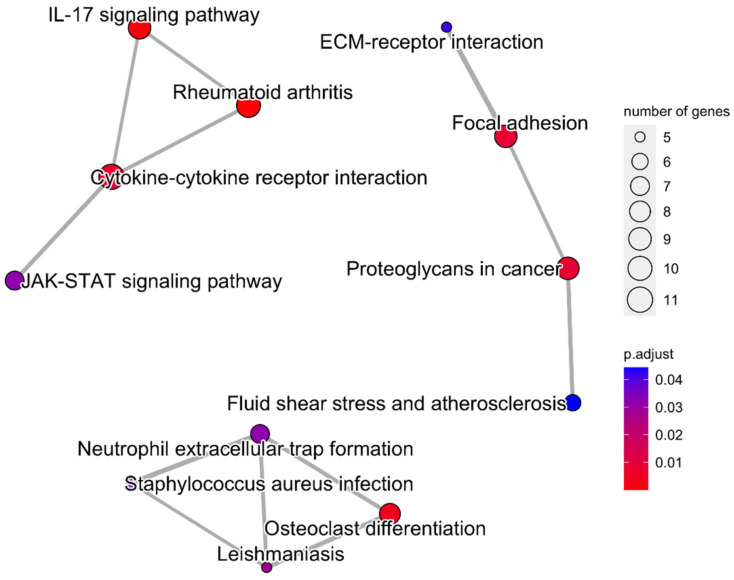
The interaction between the top five pathways associated with anti-TNF therapy resistance in Crohn’s disease visualised on a KEGG pathway diagram.

**Figure 4 jfb-13-00036-f004:**
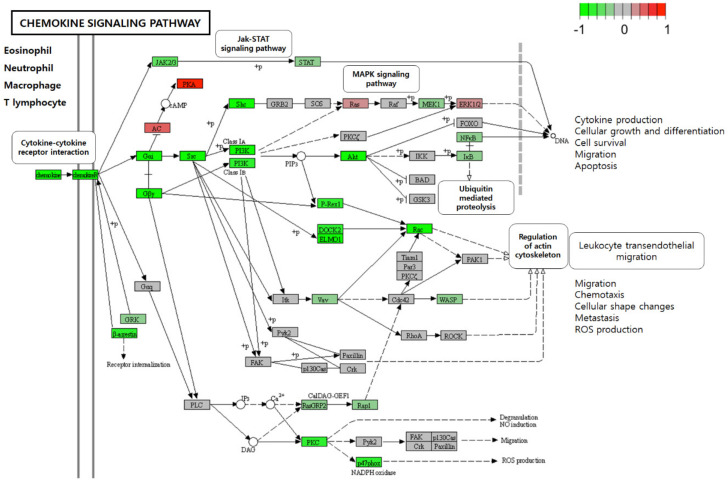
Expression changes of target genes in the chemokine signaling pathway associated with anti-TNF therapy resistance in Crohn’s disease mapped by colours: red: statistically significant increase in expression; green: statistically significant decrease in expression; and grey: expression statistically insignificant.

**Table 1 jfb-13-00036-t001:** Cellular landscape of the immune microenvironment in the non-responders versus responders to the anti-TNF therapy in Crohn’s disease.

Immune Marker	Log FC	Average Expression	t-Statistic	*p*-Value	Adjusted *p*-Value	B-Statistic
Fibroblast	−1.233	9.062	−4.284	<0.001	0.001	1.170
Neutrophil	−0.758	7.115	−4.052	<0.001	0.001	0.486
Endothelial cell	−0.568	5.94	−3.452	0.001	0.005	−1.196
Monocyte	−0.594	7.101	−3.045	0.004	0.009	−2.252
Macrophage/Monocyte	−0.594	7.101	−3.045	0.004	0.009	−2.252
B cell	−0.476	7.407	−1.434	0.159	0.291	−5.432
NK cell	0.218	4.652	1.299	0.201	0.315	−5.608
Myeloid dendritic cell	−0.227	5.085	−1.196	0.238	0.327	−5.731
cytotoxicity score	−0.220	4.861	−0.996	0.325	0.397	−5.945
T cell CD8^+^	−0.181	0.922	−0.89	0.378	0.416	−6.042
T cell	−0.061	5.295	−0.349	0.729	0.729	−6.373

Abbreviation: FC, fold change.

**Table 2 jfb-13-00036-t002:** The expression of candidate blood biomarkers and their GeneHancer elements associated with IL-17 for response to anti-TNF treatment in Crohn’s disease.

Gene	Log FC	Average Expression	t-Statistic	*p*-Value	Adjusted *p*-Value	B-Statistic	GeneHancer Identifiers	Activity
CBR4	0.284	7.105	4.778	<0.0001	0.236	2.232	-	-
ACAD10	0.215	7.08	3.981	<0.0001	0.246	0.228	GH12J112025	Enhancer active in:
								CD4^+^ CD25^−^ IL17^−^ PMA Th
								primary cells
BAIAP3	−0.264	6.6	−4.538	<0.0001	0.246	1.618	-	-
BMP6	−0.702	7.506	−4.298	<0.0001	0.246	1.013	-	-
DDX11L2	−0.767	8.115	−4.104	<0.0001	0.246	0.532	GH02J113885	Enhancer active in:
								CD4^+^ CD25^−^ IL17^−^ PMA Th
								primary cells
DYNLL2	−0.317	8.359	−4.292	<0.0001	0.246	0.999	GH17J058328	Enhancer active in:
								CD4^+^ CD25^−^ IL17^−^ PMA Th
								primary cells, CD4^+^ CD25^−^
								IL17^+^ PMA Th17 primary cells
KAT2B	−0.587	9.99	−3.945	<0.0001	0.246	0.142	GH03J020038	Enhancer active in:
							GH03J020037	CD4^+^ CD25^−^ IL17^+^ PMA Th17
							GH03J020048	primary cells
							GH03J020047	
							GH03J020060	
							GH03J020053	
							GH03J020070	
KLF9	−0.29	7.425	−3.859	<0.0001	0.246	−0.068	GH09J070402	Enhancer active in:
							GH09J070395	CD4^+^ CD25^−^ IL17- PMA Th
							GH09J070389	primary cells, CD4^+^ CD25^−^
							GH09J070400	IL17^+^ PMA Th17 primary cells
							GH09J070394	
SPTSSB	−0.465	5.504	−4.024	<0.0001	0.246	0.335	-	-

Abbreviation: FC, fold change; PMA, phorbol-12-Myristate-13-Acetate; Th, helper T-cell.

## Data Availability

The data that support the findings of this study are derived from publicly available sources, (Gene Expression Omnibus database, http://www.ncbi.nlm.nih.gov/geo/ (accessed on 1 November 2021)).

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
