# Peer review of "Uncovering Novel Pre-Treatment Molecular Biomarkers for Anti-TNF Therapeutic Response in Patients with Crohn’s Disease"

_jfb, 2022, doi:10.3390/jfb13020036_

Round 1

Reviewer 1 Report

This work identifies differentially expressed genes in patients diagnosed with Crohn's disease and establishes that IL-17 activity in affected tissues and KAT2B expression in PBMCs are biomarkers that can help predict which patients will respond to anti-TNF therapy prior to treatment. This work is well designed and performed, and uses previous public datasets. The conclusions are well supported, although they should be taken with caution as they are absolutely in-silico.

Minor suggestions:

There are works that have previously tried to identify DEGs in CD. It would be necessary to mention them and highlight the differences with this study.
There are also papers on gene expression studies of anti-TNF response in children with inflammatory bowel disease. They should be mentioned in the discussion and compared with their results. 

Author Response

<Reviewer comments>

Comments to the Author

<reviewer 1>

This work identifies differentially expressed genes in patients diagnosed with Crohn's disease and establishes that IL-17 activity in affected tissues and KAT2B expression in PBMCs are biomarkers that can help predict which patients will respond to anti-TNF therapy prior to treatment. This work is well designed and performed, and uses previous public datasets. The conclusions are well supported, although they should be taken with caution as they are absolutely in-silico.

Response: Thank you very much for your kind comments and detailed review about our study.

Minor suggestions:

There are works that have previously tried to identify DEGs in CD. It would be necessary to mention them and highlight the differences with this study.

There are also papers on gene expression studies of anti-TNF response in children with inflammatory bowel disease. They should be mentioned in the discussion and compared with their results.

Response: Thank you for the comment and we agree. Accordingly, we have edited the DISCUSSION section as the reviewer’s comment (Page 9); “Only a few studies have used this approach, although the identification of transcriptomic biomarkers using DEGs in CD could be useful to elucidate the mechanism of the phenotype and to identify the patients who are less likely to respond in early stages or even before initiation for anti-TNF therapy. Unfortunately, most of these studies used either tissue or blood samples, while we use the multi-omics data from tissue and blood samples to characterize CD patients by underlying gene transcription signatures and identify DEGs impacting on biological pathways in anti-TNF resistance.”

Reviewer 2 Report

Reviewer comments:

Comments to the Author

The authors have utilized the transcriptome analyses to predict anti-TNF therapy response in Crohn’s disease patients using data from the Gene Expression Omnibus, GeneCards, and Human Protein Atlas databases. They identified several pathways associated with the anti-TNF therapy resistance in CD patients encompassed immune pathways, including Interleukin-17 (IL-17) signaling, cytokine-cytokine receptor interaction, and rheumatoid arthritis. Moreover, authors described the anti-TNF therapy resistance in CD patients may be attributed to the low immune cell activation due to differential gene expression status of IL-17A/IL-17F between the responders and non-responders to anti-TNF agents. Interestingly, microenvironmental cell profile analysis revealed significantly lower scores of several immune-cell subsets; neutrophils, monocytes, and macrophage/monocyte in anti-TNF therapy resistant patients compared to those in the CD patients that responded to the drug.

This manuscript is very interesting and suggesting KAT2B gene as the biomarkers for IL-17 cytokine mediated neutrophil mobilization and activation in anti-TNF therapy resistant CD patients. However, the only limitation to this manuscript is the lack of their findings in biological models. Although, the manuscript for most part well written except some places in the methodological section.

Major criticisms

  • Please describe how the Microenvironmental Feature Evaluation was performed.
  • Authors need to discuss about other biomarkers available for the assessment of Crohn’s disease such as Serum levels of anti-TNF mAbs, serum levels of C reactive proteins and anti-drug antibody. Also, explain why their selected biomarkers are better than the above mentioned.
  • Further detail mechanism to show the involvement of KAT2B gene, associated with IL-17 cytokine mediated neutrophil mobilization and activation in anti-TNF therapy resistant CD patients must be elaborated using a simplified diagram. This study needs to confirm their data via other methods at the translational and transcriptional levels to impactfully pose its findings.
  • There is no biological experiment result to show the direct relevance of KAT2B gene association with IL-17 cytokine mediated neutrophil mobilization and activation in anti-TNF therapy resistant CD. Please perform some experiment to evaluate the role of KAT2B gene or protein by overexpressing its level in some biological models to confirm its role in target-based therapy in relevance to anti-TNF therapy resistant CD patients.

Minor criticisms

  • Please undergo a thorough check of the manuscript for typographical and grammatical errors.

Author Response

<Reviewer comments>

Comments to the Author

<reviewer 2>

The authors have utilized the transcriptome analyses to predict anti-TNF therapy response in Crohn’s disease patients using data from the Gene Expression Omnibus, GeneCards, and Human Protein Atlas databases. They identified several pathways associated with the anti-TNF therapy resistance in CD patients encompassed immune pathways, including Interleukin-17 (IL-17) signaling, cytokine-cytokine receptor interaction, and rheumatoid arthritis. Moreover, authors described the anti-TNF therapy resistance in CD patients may be attributed to the low immune cell activation due to differential gene expression status of IL-17A/IL-17F between the responders and non-responders to anti-TNF agents. Interestingly, microenvironmental cell profile analysis revealed significantly lower scores of several immune-cell subsets; neutrophils, monocytes, and macrophage/monocyte in anti-TNF therapy resistant patients compared to those in the CD patients that responded to the drug.

This manuscript is very interesting and suggesting KAT2B gene as the biomarkers for IL-17 cytokine mediated neutrophil mobilization and activation in anti-TNF therapy resistant CD patients. However, the only limitation to this manuscript is the lack of their findings in biological models. Although, the manuscript for most part well written except some places in the methodological section.

Response: Thank you very much for your kind comments and detailed review about our study.

Major criticisms

Please describe how the Microenvironmental Feature Evaluation was performed.

Response: Thank you for the good comment. We already mentioned the method for the immune microenvironment evaluation; however, we have added the information of the method in more detail as the reviewer’s comment. (Page 3); “The signatures were discovered using the specific cut-offs through transcriptomic analysis and underwent an additional selection process. A sub-signature with strong inter-marker correlation was kept following hierarchical clustering of the whole signature transcriptomic markers. Then, the in silico simulated mixtures were computed as follows: firstly, weights for all included populations were chosen randomly. Pure transcriptomic profiles for all populations were computed with the expression of all genes being the mean expression over all the corresponding samples in the datasets.”

Thank you for your detailed review, again.

Authors need to discuss about other biomarkers available for the assessment of Crohn’s disease such asSerum levels of anti-TNF mAbs, serum levels of C reactive proteins and anti-drug antibody. Also, explain why their selected biomarkers are better than the above mentioned.

Response: Thank you for this detailed comment. We have added the sentences as the reviewer’s comment (Page 9); “. Several biologic markers such as serum cytokine levels (TNF and IL-6), C-reactive protein, anti-neutrophil cytoplasmic antibody, anti-Saccharomyces cerevisiae antibody and anti-drug antibody have been studied. However, they could not unmask the mechanisms represented by the nature of observational studies, we, therefore, have not yet seen data convincing enough to use this routinely in clinical practice.”

Further detail mechanism to show the involvement of KAT2B gene, associated with IL-17 cytokine mediated neutrophil mobilization and activation in anti-TNF therapy resistant CD patients must be elaborated using a simplified diagram. This study needs to confirm their data via other methods at the translational and transcriptional levels to impactfully pose its findings.

Response: Thank you very much for your kind comments and detailed review about our study. We agree that, as you have highlighted, the issue regarding the further study is a very important point.

As we mentioned in limitation section, few databases with multi-omics data from one sample in CD are existed.

Therefore, we planned a multicenter study by the Group of the Korean Association for the Study of Intestinal Disease (KASID) to confirm our results.

Thank you for this good comment, again.

There is no biological experiment result to show the direct relevance of KAT2B gene association with IL-17 cytokine mediated neutrophil mobilization and activation in anti-TNF therapy resistant CD. Please perform some experiment to evaluate the role of KAT2B gene or protein by overexpressing its level insome biological models to confirm its role in target-based therapy in relevance to anti-TNF therapy resistant CD patients.

Response: Thank you very much for your kind comments and detailed review about our study. We completely agree with the reviewer’s comment.

Therefore, we will obtain the tissue samples in CD patients from the KASID multicenter study and perform the experimental validation of our results.

Please excuse us.

Minor criticisms

Please undergo a thorough check of the manuscript for typographical and grammatical errors.

Response: Thank you for the comment. Accordingly, we have edited the manuscript.

This manuscript is a resubmission of an earlier submission. The following is a list of the peer review reports and author responses from that submission.

Round 1

Reviewer 1 Report

   In this manuscript, Kwak MS and colleagues have searched for novel molecular biomarkers for Anti-TNF therapeutic response using datasets from gene expression profiles from serum and tissue samples from Crohn’s disease patients. This manuscript is focused on a hot topic because anti-TNF therapy has led to marked improvements in the treatment of patients with Crohn’s disease, but approximately one-third of patients do not respond to anti-TNF induction therapy, highlighting the need to identify biomarkers for therapeutic response in Crohn’s disease.

   Interestingly, they found that in the intestinal tissues of Crohn’s disease patients, IL17 signaling turned out in a diverse immune microenvironment which in turn, promotes anti-TNF drug resistance. Furthermore, they found that KAT2B gene expression in PBMCs of Crohn’s disease patients could be a prognostic and immunologic biomarker for anti-TNF resistance.

   However, the study is based only on silico studies, without experimental validation, and more importantly, the authors did not consider other factors that may be influencing the analysis of the data which detracts from the value of the study as detailed below.

General comments:

First, the authors analyzed the gene expression profiles from ileal specimens of 36 CD patients (GSE16879) including information about CD therapy response from 19 responders and 17 non-responders to anti-TNF drugs. Several critical points have been observed in this analysis.

  1. In the original article, there are colonic and ileal samples, but the authors described the cohort as ileal specimens of 36 CD patients. Please correct it.
  2. In the original article, the authors classified the patients in 20 Responders and 17 non-responders to IFX treatment. Why did the authors delete one responder patient? Please, justify.
  3. As the authors are searching for molecular biomarkers for Anti-TNF therapeutic response in patients with Crohn’s disease, it is necessary to include some clinical data from the cohort which may influence the results. Have the authors considered other clinical variables that affect your analysis such as disease location (ileal versus colonic), other medication such as corticosteroids, methotrexate, …, smoking? Please, comment on that.
  4. The authors should include one table with clinical variables used in the two groups analyzed Responders versus primary Non-Responders where the readers can easily know the anti-TNF used (in this case infliximab), doses, week or weeks where the evaluation and classification as Responder or Non-Responder where performed, other treatments…
  5. Could the authors add a figure to show the functional enrichment analysis based on down-regulated genes and other based on up-regulated genes? It would be interesting for the readers to better understand the genes associated with anti-TNF resistance in Crohn’s disease.

   Second, the authors analyzed the microarray expression data from peripheral blood mononuclear cells (PBMCs) in a total of 14 responders and 6 primary non-responders to anti-TNF therapy. In this case, the cohort was not modified from the original paper which the readers may assume no significant differences between groups.

  1. The main objective of the work where the data was obtained (GSE42296) is to find gene panels that may differentiate Responders from Non-Responders to Anti-TNF therapy at baseline in Crohn’s disease. Thus, the analysis performed by the authors in this cohort is a repetition.
  2. Moreover, clinical data from the cohort such as anti-TNF used, doses, weeks of evaluation, should be included in the manuscript for a better understanding the manuscript.

   Finally, it is not clearly described in the manuscript whether the microenvironmental feature evaluation was performed by the authors or was extracted from the transcriptomic data from the other papers (GSE16879?). If it was performed by the authors, from which human cohort? Clinical data of the cohort used must be described in the manuscript. If not, please described in the Methods section adequately.

Author Response

Pharmaceutics

Dear Editors-in-Chief,

We appreciate the opportunity to revise our work for consideration for publication in Pharmaceutics. We hope that our revisions are to your satisfaction. We present below our detailed responses to each of the reviewers’ concerns and comments.

We edited the manuscript by professional English-editing service (Please see the certification).

And, we added the co-author (Su Bee Park) who revised the manuscript finally.

<reviewer 1>

In this manuscript, Kwak MS and colleagues have searched for novel molecular biomarkers for Anti-TNF therapeutic response using datasets from gene expression profiles from serum and tissue samples from Crohn’s disease patients. This manuscript is focused on a hot topic because anti-TNF therapy has led to marked improvements in the treatment of patients with Crohn’s disease, but approximately one-third of patients do not respond to anti-TNF induction therapy, highlighting the need to identify biomarkers for therapeutic response in Crohn’s disease.

   Interestingly, they found that in the intestinal tissues of Crohn’s disease patients, IL17 signaling turned out in a diverse immune microenvironment which in turn, promotes anti-TNF drug resistance. Furthermore, they found that KAT2B gene expression in PBMCs of Crohn’s disease patients could be a prognostic and immunologic biomarker for anti-TNF resistance.

   However, the study is based only on silico studies, without experimental validation, and more importantly, the authors did not consider other factors that may be influencing the analysis of the data which detracts from the value of the study as detailed below.

General comments:

First, the authors analyzed the gene expression profiles from ileal specimens of 36 CD patients (GSE16879) including information about CD therapy response from 19 responders and 17 non-responders to anti-TNF drugs. Several critical points have been observed in this analysis.

Response: Thank you very much for your kind comments and detailed review about our study.

In the original article, there are colonic and ileal samples, but the authors described the cohort as ileal specimens of 36 CD patients. Please correct it.

Response: Thank you for the good comment. We have changed it as the reviewer’s comment (Page 5); “We analysed the gene expression profiles from the ileo-colonic specimen of 36 CD patients from the GSE16879 dataset that includes information about CD therapy response from 19 responders and 17 primary non-responders for anti-TNF drugs.”

In the original article, the authors classified the patients in 20 Responders and 17 non-responders to IFX treatment. Why did the authors delete one responder patient? Please, justify.

Response: Thank you for the detailed comment. We analyzed the data after excluding the patient (GSM423031) without the information of post-treatment transcriptional data.

As the authors are searching for molecular biomarkers for Anti-TNF therapeutic response in patients with Crohn’s disease, it is necessary to include some clinical data from the cohort which may influence the results. Have the authors considered other clinical variables that affect your analysis such as disease location (ileal versus colonic), other medication such as corticosteroids, methotrexate, …, smoking? Please, comment on that.

Response: Thank you for the good comment. We completely agree with the reviewer’s comment.

We analyzed the GEO database for real-world evaluations of the anti-TNF resistance in CD using the molecular data; however, we would like to acknowledge the limitations of the present investigation because clinical information is not limited in the database. Therefore, we have described the limitation in the DISCUSSION SECTION (Page 13); “Third is the lack of clinical information, such as age, gender, smoking status, and co-prescribed drugs.”

Currently, we plan on conducting multi-center studies for improved stratified comparisons; however, it will require a lot of time.

The authors should include one table with clinical variables used in the two groups analyzed Responders versus primary Non-Responders where the readers can easily know the anti-TNF used (in this case infliximab), doses, week or weeks where the evaluation and classification as Responder or Non-Responder where performed, other treatments…

Response: Thank you for the good comment. we have added the information of data in the supplementary table 1. Some variables are not listed, because of the lack of public data sets for the drug doses and treatments.

Could the authors add a figure to show the functional enrichment analysis based on down-regulated genes and other based on up-regulated genes? It would be interesting for the readers to better understand the genes associated with anti-TNF resistance in Crohn’s disease.

Response: Thank you for the good comment. We already analyzed the data using the up- or down- regulated genes for anti-TNF resistance and illustrated the results on the figures.

Thank you for your detailed review, again.

   Second, the authors analyzed the microarray expression data from peripheral blood mononuclear cells (PBMCs) in a total of 14 responders and 6 primary non-responders to anti-TNF therapy. In this case, the cohort was not modified from the original paper which the readers may assume no significant differences between groups.

The main objective of the work where the data was obtained (GSE42296) is to find gene panels that may differentiate Responders from Non-Responders to Anti-TNF therapy at baseline in Crohn’s disease. Thus, the analysis performed by the authors in this cohort is a repetition.

Moreover, clinical data from the cohort such as anti-TNF used, doses, weeks of evaluation, should be included in the manuscript for a better understanding the manuscript.

Response: Thank you for the comment and we agree. Accordingly, we have edited the Method section as the reviewer’s comment (Page 5); “All patients received infliximab caused by the resistance to corticosteroids and/or immunosuppressants and their samples were obtained before and after therapy. Detailed information of the datasets can be found in supplementary table 1.” Also, we have added the information of data in the supplementary table 1.

   Finally, it is not clearly described in the manuscript whether the microenvironmental feature evaluation was performed by the authors or was extracted from the transcriptomic data from the other papers (GSE16879?). If it was performed by the authors, from which human cohort? Clinical data of the cohort used must be described in the manuscript. If not, please described in the Methods section adequately.

Response: Thank you very much for your kind comments and detailed review about our study. We agree that, as you have highlighted, the issue regarding the microenvironmental characteristics of immune-competent subtypes is a very important point.

We cited the original paper and added the information about the method in more detail in the Method section (Page 7, Reference 19); “This method can robustly quantify the abundance of various immune and stromal cell populations based on transcriptomic data for each sample. The output of MCP-counter can be used to estimate the relative infiltration of endothelial cells, fibroblasts, and another eight immune cells populations.”

The method is from background-prediction caused by its nature of translational estimation, and is not straightforward. However, the MCP-counter scores summarize the expression of the transcriptomic markers specific for a given population, and have been validated for their ability to correlate with the fraction of mRNA originating from the corresponding cell population and to cell infiltration estimated by immunohistochemistry.

We aimed to evaluate the mechanisms and to identify the predictive markers for anti-TNF resistance using several validated algorithms and bio-databases.

Reviewer 2 Report

The manuscript discusses the possible biomarkers to identify Crohn's disease patients who are resistant to treatment by monoclonal antibody against tumour necrosis factor (TNF). Transcriptome analysis revealed that several immune pathways were less active and a specific gene, KAT2B, was highly under-expressed in these patients. Overall, downregulation of IL-17 signaling pathway emerges to be the reason of why these patients were resistant to anti-TNF treatment. 

Overall, I think this is a very good manuscript. I have nothing constructive to add that will make the analysis stronger. I suggest quick publication and congratulate all the authors. 

Author Response

Pharmaceutics

Dear Editors-in-Chief,

We appreciate the opportunity to revise our work for consideration for publication in Pharmaceutics. We hope that our revisions are to your satisfaction. We present below our detailed responses to each of the reviewers’ concerns and comments.

We edited the manuscript by professional English-editing service (Please see the certification).

And, we added the co-author (Su Bee Park) who revised the manuscript finally.

<reviewer 2>

The manuscript discusses the possible biomarkers to identify Crohn's disease patients who are resistant to treatment by monoclonal antibody against tumour necrosis factor (TNF). Transcriptome analysis revealed that several immune pathways were less active and a specific gene, KAT2B, was highly under-expressed in these patients. Overall, downregulation of IL-17 signaling pathway emerges to be the reason of why these patients were resistant to anti-TNF treatment.

Overall, I think this is a very good manuscript. I have nothing constructive to add that will make the analysis stronger. I suggest quick publication and congratulate all the authors.

Response: Thank you very much for your kind comments and detailed review about our study.

Reviewer 3 Report

The study "Uncovering novel pre-treatment molecular biomarkers for anti-tnf therapeutic response in patients with Chron's disease" by Kwak et al uses an interesting approach trying to elucidate molecular mechanisms responsible for anti-TNF therapy resistance in CD patients using transcriptomics.

The study is very interesting and reveals interesting pathways that differ between responders and non-responders. The major drawback is that the authors have not verified their findings in patient samples.

I have some comments and concerns:

1) Verification of the findings in biological samples from CD patients of the findings has not been performed. This would strengthen the paper considerably.

2) Discussion on the contribution of transmembrane and soluble TNF to CD pathology should be included, just as the relevance of TNFR1 and TNFR2 levels in CD patients. solTNFR1 and solTNFR2 act as biological inhibitors of solTNF and especially solTNF-TNFR1 signaling is known to mediate detrimental inflammatory effects, whereas tmTNF-TNFR2 signaling is known to be important for innate immunity. A thorough discussion on this and the data in the literature on CD patients, and possibly also other patient types with chronic inflammatory diseases, is warranted.

3) The figures should be presented consecutively. Figure 4 is presented before Figure 3 in the current manuscript.

4) Figure 4 is hard to read, please enlarge the pathways or at least the size of the letters.

Author Response

Pharmaceutics

Dear Editors-in-Chief,

We appreciate the opportunity to revise our work for consideration for publication in Pharmaceutics. We hope that our revisions are to your satisfaction. We present below our detailed responses to each of the reviewers’ concerns and comments.

We edited the manuscript by professional English-editing service (Please see the certification).

And, we added the co-author (Su Bee Park) who revised the manuscript finally.

<reviewer 3>

The study "Uncovering novel pre-treatment molecular biomarkers for anti-tnf therapeutic response in patients with Chron's disease" by Kwak et al uses an interesting approach trying to elucidate molecular mechanisms responsible for anti-TNF therapy resistance in CD patients using transcriptomics.

The study is very interesting and reveals interesting pathways that differ between responders and non-responders. The major drawback is that the authors have not verified their findings in patient samples.

Response: Thank you very much for your kind comments and detailed review about our study.

I have some comments and concerns:

1) Verification of the findings in biological samples from CD patients of the findings has not been performed. This would strengthen the paper considerably.

Response: Thank you for highlighting this point. Therefore, we planned a multicenter study by the Group of the Korean Association for the Study of Intestinal Disease (KASID) to confirm our results.

2) Discussion on the contribution of transmembrane and soluble TNF to CD pathology should be included, just as the relevance of TNFR1 and TNFR2 levels in CD patients. solTNFR1 and solTNFR2 act as biological inhibitors of solTNF and especially solTNF-TNFR1 signaling is known to mediate detrimental inflammatory effects, whereas tmTNF-TNFR2 signaling is known to be important for innate immunity. A thorough discussion on this and the data in the literature on CD patients, and possibly also other patient types with chronic inflammatory diseases, is warranted.

3) The figures should be presented consecutively. Figure 4 is presented before Figure 3 in the current manuscript.

Response: Thank you for the comment and we agree. We will contact the editorial team for correction.

4) Figure 4 is hard to read, please enlarge the pathways or at least the size of the letters.

Response: Thank you for the comment.

We tried our best to edit the figure, initially.

However, the overall pathway is vast, therefore we increased the font size of pathways or cell names as much as possible. (..so as not to interfere with the overall picture)

Thank you for your detailed comment, again.

Round 2

Reviewer 1 Report

Dear Authors,

The quality of the manuscript has significantly improved.

Reviewer 3 Report

I only see that the authors improved the English and increased the font size. I don't see any of the other comments being incorporated.